

# Ground-state energy
# of a Richardson-Gaudin integrable BCS model

Yibing Shen, Phillip S. Isaac and Jon Links⋆

School of Mathematics and Physics, The University of Queensland, 4072 Australia

⋆ jrl@maths.uq.edu.au

## Abstract

We investigate the ground-state energy of a Richardson-Gaudin integrable BCS model, generalizing the closed and open $p + ip$ models. The Hamiltonian supports a family of mutually commuting conserved operators satisfying quadratic relations. From the eigenvalues of the conserved operators we derive, in the continuum limit, an integral equation for which a solution corresponding to the ground state is established. The energy expression from this solution agrees with the BCS mean-field result.

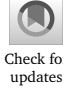
## 1 Introduction

The $p_x + ip_y$-pairing Hamiltonian is an integrable example [1] of Bardeen-Cooper-Schrieffer (BCS) superconducting model [2]. It is a chiral variant of the $p+ip$-pairing model [3]. Like its

better known ancestor the *s*-wave pairing model [2] which is of Richardson-Gaudin type [4–6], the $p_x + i p_y$ BCS Hamiltonian also supports a family of mutually commuting operators [7]. The Bethe Ansatz solution of the $p_x + i p_y$-pairing model was initially established with conserved particle number and no interaction with the environment [1,8,9]. For this reason, we call the integrable $p_x + i p_y$ BCS model the 'closed' $p + i p$ model. An extension of the closed model was later studied where the model is coupled to its environment. This extended model no longer conserves particle number, and thus cannot be considered as describing a closed system. Therefore we refer to it as an open model. The interaction term corresponding to particle exchange with the system's environment leads to $u(1)$ symmetry being broken, which generally renders the analysis of the system to be more complicated. Integrability of the open model was established in [10] through use of the boundary Quantum Inverse Scattering Method. An alternative derivation, which is less technical, was later provided in [11].

The $p_x + i p_y$-wave interaction also gives rise to non-trivial topological phase transitions by breaking time-reversal symmetry [12,13]. Recently, there has been much attention to the topological properties of this model using various methods. In particular, the exact Bethe Ansatz solutions provide insight into the topological behaviour of the system through the peculiar behaviour of Bethe roots [7,9,14]. Topological properties of the open model have been studied in [15,16].

In the recent work by Skrypnyk [17], a generalization of the closed and open $p + i p$ models was constructed by considering a non-skew-symmetric $r$-matrix satisfying a generalized classical Yang-Baxter equation. The underlying conserved operators of this generalized model were also independently constructed by Claeys et al [18], as a spin-1/2 Richardson-Gaudin XYZ solution where a non-zero magnetic field component is present. This generalized model opens up an avenue to investigate more complicated interaction with the environment.

Originating in the work of Babelon and Talalaev [19], it was shown, through a change of variables, that the Bethe Ansatz Equations (BAE) for certain Richardson-Gaudin type systems can be recast into a set of coupled polynomial equations. The roots of these equations can be expressed in the form of eigenvalues of the self-adjoint conserved operators, and consequently are real-valued. The same form of polynomial equations were adopted in [20] as means for efficient numerical solution of the conserved operator spectrum, and in [21] to compute wavefunction overlaps. This technique was shown to be generalizable in [22,23]. It was subsequently shown for some systems that the conserved operator eigenvalues (COE), satisfying quadratic relations, are inherited from the same relations at the operator level [15,24]. The generalized model considered below also shares this property of underlying conserved operators satisfying quadratic relations [17,18]. However the polynomial equations for the generalized model, leading to the BAE, remain unknown.

The conventional Bethe Ansatz analysis of Richardson-Gaudin models in the thermodynamic limit suffers complications in that one needs to solve for a density function on an unknown curve in the complex plane which captures the behaviors of the Bethe roots [25–27]. For the closed model, the complex curve and its density were studied extensively in [9] for different regions of the phase diagram, with the resulting ground-state energies shown to be consistent with mean-field analysis. However, limitations of this method remain and are discussed in [28]. For the open model, even more intricate behavior of the Bethe roots is revealed from numerical analysis which renders it a difficult task to capture the Bethe roots with a piecewise continuous complex curve. Alternatively, we proposed a new method [28] for deriving the continuum limit ground-state energy in terms of the COE, initially for the closed model. This method does not rely on any hypothesis on the distribution of Bethe roots, hence its validity extends to regions of the phase diagram where the Bethe root evolution is not well-described by a complex curve. This new method was also shown to accommodate the open model.

The objective of our study is to apply this new approach to derive the ground-state energy

of the generalized model in the continuum limit. The main focus of the task is solving the integral equation arising from the quadratic relations of the COE in the continuum limit. The proof of the solution will be given in detail, which is a generalization of the similar approach for the open case. Here it will also be shown that the result is in agreement with mean-field calculations.

The generalized integrable BCS Richardson-Gaudin Hamiltonian is introduced in Sect. 2 with its underlying conserved operators satisfying quadratic relations. In Sect. 3, results from a BCS mean-field analysis of the generalized model, including the ground-state energy, are given. In Sect. 4, attention turns towards using the alternative approach to derive the ground-state energy, assisted by establishing a solution to the integral equation which corresponds to the ground state of the model. Concluding remarks and discussion are offered in Sect. 5.

## 2 The Hamiltonian

Consider a system of fermions and let $c_{\alpha j}$, $c_{\alpha' k}^\dagger$, $\alpha, \alpha' \in \{+, -\}$, $j, k \in 1, 2, \ldots, L$ denote the annihilation and creation operators, satisfying

$$\{c_{\alpha j}, c_{\alpha' k}\} = \{c_{\alpha j}^\dagger, c_{\alpha' k}^\dagger\} = 0, \qquad \{c_{\alpha j}^\dagger, c_{\alpha' k}\} = \delta_{\alpha \alpha'} \delta_{jk}.$$

We consider the following integrable BCS Richardson-Gaudin Hamiltonian reported in [17],

$$H_{BCS} = \frac{1}{2} \sum_{i=1}^L f_i^+ f_i^- (c_{+i}^\dagger c_{+i} + c_{-i}^\dagger c_{-i}) + \frac{g}{2} \sum_{i=1}^L \sum_{j \neq i}^L (f_i^- f_j^- + f_i^+ f_j^+) c_{+i}^\dagger c_{-i}^\dagger c_{-j} c_{+j}$$

$$+ \sum_{i=1}^L (f_i^- \gamma + i f_i^+ \lambda) c_{-i} c_{+i} + \sum_{i=1}^L (f_i^- \gamma - i f_i^+ \lambda) c_{+i}^\dagger c_{-i}^\dagger$$

$$+ \frac{g}{4} \sum_{i=1}^L \sum_{j \neq i}^L (f_i^- f_j^- - f_i^+ f_j^+)(c_{+i}^\dagger c_{-i}^\dagger c_{+j}^\dagger c_{-j}^\dagger + c_{-i} c_{+i} c_{-j} c_{+j}), \tag{1}$$

where $g, \gamma, \lambda \in \mathbb{R}$ and

$$f_k^+ = \sqrt{\epsilon_k + \beta_x}, \quad f_k^- = \sqrt{\epsilon_k + \beta_y}, \tag{2}$$

with $\epsilon_k > 0$, $\epsilon_k + \beta_x > 0$ and $\epsilon_k + \beta_y > 0$ for all $k = 1, \ldots, L$. Assuming $\beta_x \geq \beta_y$ and setting

$$z_k^2 = \epsilon_k + \frac{\beta_x + \beta_y}{2}, \quad \beta = \frac{\beta_x - \beta_y}{2},$$

we have

$$f_k^+ = \sqrt{z_k^2 + \beta}, \quad f_k^- = \sqrt{z_k^2 - \beta}.$$

The Hamiltonian has the form of a standard BCS model, (the terms in the first line of (1)), with additional terms. The standard terms are associated with the single-particle energy spectrum and the scattering of Cooper pairs. The additional terms are responsible for the breaking of the $u(1)$-symmetry associated to the total particle number, and are interpreted as interaction with the system's environment. The Hamiltonian (1) is a generalisation of the open $p + ip$ Hamiltonian [10]. By setting $\lambda = 0$ and

$$\beta_x = \beta_y \iff \beta = 0,$$

we recover the conserved operators underlying the open $p + ip$ model. Furthermore, if we also set $\gamma = 0$ these reduce to those of the closed $p + ip$ model.

Now we replace the fermion operators with the generators $S_k^+$, $S_k^-$ and $S_k^z$ of the $sl(2)^{\otimes L}$ algebra, through

$$S_k^+ = c_{+k}^\dagger c_{-k}^\dagger, \quad S_k^- = c_{-k}c_{+k}, \quad S_k^z = \frac{1}{2}(c_{+k}^\dagger c_{+k} + c_{-k}^\dagger c_{-k} - I),$$

where $k = 1, \ldots, L$. These operators satisfy the following commutation relations

$$[S_k^z, S_k^\pm] = \pm S_k^\pm, \quad [S_k^+, S_k^-] = 2S_k^z.$$

The Hamiltonian $H_{BCS}$ can be rewritten (omitting the constant term) in the following form

$$H = \sum_{i=1}^{L} f_i^+ f_i^- S_i^z + \sum_{i=1}^{L} \left[ \left(f_i^- \gamma - i f_i^+ \lambda\right) S_i^+ + \left(f_i^- \gamma + i f_i^+ \lambda\right) S_i^- \right]$$
$$+ \frac{g}{2} \sum_{i=1}^{L} \sum_{j \neq i}^{L} (f_i^- f_j^- + f_i^+ f_j^+) S_i^+ S_j^- + \frac{g}{4} \sum_{i=1}^{L} \sum_{j \neq i}^{L} (f_i^- f_j^- - f_i^+ f_j^+)(S_i^+ S_j^+ + S_i^- S_j^-). \quad (3)$$

Define the following operators

$$Q_i = S_i^z + \frac{2\gamma}{f_i^+} S_i^x + \frac{2\lambda}{f_i^-} S_i^y + 2g \sum_{j \neq i}^{L} \frac{1}{z_i^2 - z_j^2} \left(f_i^+ f_j^- S_i^x S_j^x + f_i^- f_j^+ S_i^y S_j^y\right)$$
$$+ 2g \sum_{j \neq i}^{L} \frac{f_j^+ f_j^-}{z_i^2 - z_j^2} \left(S_i^z S_j^z - \frac{1}{4}\right), \quad (4)$$

with $S_i^x$, $S_i^y$ given by

$$S_i^x = \frac{1}{2}(S_i^+ + S_i^-), \quad S_i^y = -\frac{i}{2}(S_i^+ - S_i^-).$$

It can be shown that

$$H = \sum_{i=1}^{L} f_i^+ f_i^- Q_i,$$

where $\{Q_i\}$ is a set of mutually commuting conserved operators. These operators have been shown [17, 18] to satisfy the following quadratic relations:

$$Q_i^2 = \frac{I}{4} + \frac{\gamma^2}{(f_i^+)^2} + \frac{\lambda^2}{(f_i^-)^2} - g \sum_{j \neq i}^{L} f_j^+ f_j^- \left(\frac{Q_i - Q_j}{z_i^2 - z_j^2}\right) + g^2 \beta^2 \sum_{j \neq i}^{L} \left(\frac{1}{f_i^+ f_j^- + f_i^- f_j^+}\right)^2. \quad (5)$$

The eigenvalues $\{q_i\}$ corresponding to $\{Q_i\}$ give the energy expression

$$E = \sum_{i=1}^{L} f_i^+ f_i^- q_i, \quad (6)$$

and due to (5), satisfy the relation

$$q_i^2 = \frac{1}{4} + \frac{\gamma^2}{(f_i^+)^2} + \frac{\lambda^2}{(f_i^-)^2} - g \sum_{j \neq i}^{L} f_j^+ f_j^- \left(\frac{q_i - q_j}{z_i^2 - z_j^2}\right) + g^2 \beta^2 \sum_{j \neq i}^{L} \left(\frac{1}{f_i^+ f_j^- + f_i^- f_j^+}\right)^2. \quad (7)$$

## 3 Mean-field analysis

The Hamiltonian $H$ in (3) can be rewritten as

$$H = \sum_{i=1}^{L} f_i^+ f_i^- S_i^z + 2\sum_{i=1}^{L}\left(\gamma f_i^- S_i^x + \lambda f_i^+ S_i^y\right)$$
$$+ g\sum_{i=1}^{L}\sum_{j\neq i}^{L} f_i^- f_j^- S_i^x S_j^x + g\sum_{i=1}^{L}\sum_{j\neq i}^{L} f_i^+ f_j^+ S_i^y S_j^y. \tag{8}$$

Introducing the parameter

$$G = -g$$

in the following calculation, we proceed with the mean-field analysis of the model. The techniques below extend the original methods for the $s$-wave BCS model [2]. First let

$$\mathcal{S}^x = \sum_{i=1}^{L} f_i^- S_i^x, \quad \mathcal{S}^y = \sum_{i=1}^{L} f_i^+ S_i^y,$$
$$\Delta_x = 2\langle \mathcal{S}^x\rangle, \quad \Delta_y = 2\langle \mathcal{S}^y\rangle,$$

where $\Delta_x, \Delta_y \in \mathbb{R}$. Then linearizing (8) through

$$AB \approx \langle A\rangle B + A\langle B\rangle - \langle A\rangle\langle B\rangle,$$

we have

$$H \approx \sum_{i=1}^{L} f_i^+ f_i^- S_i^z + 2\gamma\mathcal{S}^x + 2\lambda\mathcal{S}^y - G\Delta_x\mathcal{S}^x - G\Delta_y\mathcal{S}^y + C,$$

where

$$C = \frac{G(\Delta_x)^2}{4} + \frac{G(\Delta_y)^2}{4} + \sum_{i=1}^{L} \frac{Gz_i^2}{2}.$$

The detailed calculations are given in Appendix A. The ground-state energy reads

$$E = -\frac{1}{2}\sum_{i=1}^{L}\sqrt{(f_i^+)^2(f_i^-)^2 + (2\gamma - G\Delta_x)^2(f_i^-)^2 + (2\lambda - G\Delta_y)^2(f_i^+)^2} + C, \tag{9}$$

subject to the 'gap' equations

$$\frac{\Delta_x}{G\Delta_x - 2\gamma} = \sum_{i=1}^{L} \frac{(f_i^-)^2}{\sqrt{(f_i^+)^2(f_i^-)^2 + (2\gamma - G\Delta_x)^2(f_i^-)^2 + (2\lambda - G\Delta_y)^2(f_i^+)^2}}, \tag{10}$$

$$\frac{\Delta_y}{G\Delta_y - 2\lambda} = \sum_{i=1}^{L} \frac{(f_i^+)^2}{\sqrt{(f_i^+)^2(f_i^-)^2 + (2\gamma - G\Delta_x)^2(f_i^-)^2 + (2\lambda - G\Delta_y)^2(f_i^+)^2}}. \tag{11}$$

That is, given $G$, $\gamma$ and $\lambda$ the two equations (10) and (11) determine the parameters $\Delta_x$ and $\Delta_y$.

It is worthwhile to point out that the open $p+ip$ model as a special case of the generalized model can be rewritten in the following form

$$H_{open} = \sum_{i=1}^{L} z_i^2 S_i^z - G \sum_{i=1}^{L} \sum_{j \neq i}^{L} z_i z_j \left( S_i^x S_j^x + S_i^y S_j^y \right) + 2\gamma \sum_{i=1}^{L} z_i S_i^x. \tag{12}$$

The 'gap' equations for the open model read

$$\Delta_x = (G\Delta_x - 2\gamma) \sum_{i=1}^{L} \frac{z_i^2}{\sqrt{z_i^4 + (G\Delta_x - 2\gamma)^2 z_i^2 + (G\Delta_y)^2 z_i^2}},$$

$$\Delta_y = (G\Delta_y) \sum_{i=1}^{L} \frac{z_i^2}{\sqrt{z_i^4 + (G\Delta_x - 2\gamma)^2 z_i^2 + (G\Delta_y)^2 z_i^2}}.$$

If we require a non-zero $\gamma$, it is straightforward from these two equations to conclude that $\Delta_y = 0$, i.e. we have one 'gap' equation

$$\frac{\Delta_x}{G\Delta_x - 2\gamma} = \sum_{i=1}^{L} \frac{z_i^2}{\sqrt{z_i^4 + (G\Delta_x - 2\gamma)^2 z_i^2}}.$$

In this case, we recover the same mean-field results, i.e. energies and gap equations, as in [28] for the open model.

Next we will introduce an integral approximation to show that the general mean-field ground-state energy (9) is consistent with the exact solution. The techniques used require an extension of those in [28], to account for the two gap equations.

## 4  Integral approximation of the quadratic identities

Recall that for an arbitrary function $F(x)$, we may consider the following integral approximation (or continuum limit) of a summation,

$$\frac{1}{L} \sum_{i=1}^{L} F(x_i) \approx \int_a^b dx\, \rho(x) F(x),$$

where $a, b$ are the lower and upper bounds for the set of real numbers $\{a = x_1 < x_2 < \cdots < x_L = b\}$ and $\rho(x)$ is a density function introduced for the distribution of $x_i$ such that

$$\rho(x_i) = \frac{1}{(L-1)(x_{i+1} - x_i)}, \quad i = 1, 2, \ldots, L-1,$$

satisfying the normalization condition

$$\int_a^b dx\, \rho(x) = 1.$$

In order to proceed with the integral approximation of (7), we assume that

$$\frac{1}{L} \sum_{j \neq i}^{L} f_j^+ f_j^- \frac{q_i - q_j}{z_i^2 - z_j^2}, \quad \frac{1}{L} \sum_{j \neq i}^{L} \left( \frac{1}{f_i^+ f_j^- + f_i^- f_j^+} \right)^2$$

approach finite limits as $L \to \infty$. We will then require the parameter $G \sim 1/L$, such that $GL$ remains finite. The last term in (7) then vanishes as $L \to \infty$ since $G^2\beta^2 L \sim 1/L$. We will adopt the notation $\mathcal{G} = GL$ in the following calculations.

The Hamiltonian (1) is a function of many independent variables, including $\{z_i\}$. It is physically plausible to interpret these particular variables as the momentum spectrum in the non-interacting limit with $\beta = 0$. Consider a density $\rho$ for the distribution of the variables $\{z_i^2\}$. This density is associated with the kinetic energies in the non-interacting limit with $\beta = 0$. The continuum limit for the quadratic identity (7) then becomes

$$q(\varepsilon)^2 = \frac{1}{4} + \frac{\gamma^2}{f^+(\varepsilon)^2} + \frac{\lambda^2}{f^-(\varepsilon)^2} + \mathcal{G}\int_{\omega_0}^{\omega} du\, \rho(u) f^+(u) f^-(u) \frac{q(\varepsilon) - q(u)}{\varepsilon - u}, \tag{13}$$

where $\omega$, $\omega_0 > \beta$ are the upper and lower bounds introduced for the continuous variable $\varepsilon$ replacing $\{z_i^2\}$ in the continuum limit, and

$$f^+(\varepsilon) = \sqrt{\varepsilon + \beta}, \qquad f^-(\varepsilon) = \sqrt{\varepsilon - \beta}.$$

Assuming that $\Delta_x, \Delta_y \sim L$, the continuum limit for the gap equations (10) and (11) read

$$\frac{\hat{\Delta}_x}{\mathcal{G}\hat{\Delta}_x - 2\gamma} = \int_{\omega_0}^{\omega} d\varepsilon\, \rho(\varepsilon) \frac{\varepsilon - \beta}{R(\varepsilon)}, \quad \frac{\hat{\Delta}_y}{\mathcal{G}\hat{\Delta}_y - 2\lambda} = \int_{\omega_0}^{\omega} d\varepsilon\, \rho(\varepsilon) \frac{\varepsilon + \beta}{R(\varepsilon)}, \tag{14}$$

where

$$\hat{\Delta}_x = \Delta_x/L, \qquad \hat{\Delta}_y = \Delta_y/L,$$

and we have also set

$$R(\varepsilon) = \sqrt{\left(f^+(\varepsilon)\right)^2 \left(f^-(\varepsilon)\right)^2 + (2\gamma - \mathcal{G}\hat{\Delta}_x)^2 \left(f^-(\varepsilon)\right)^2 + (2\lambda - \mathcal{G}\hat{\Delta}_y)^2 \left(f^+(\varepsilon)\right)^2}. \tag{15}$$

## 4.1 Solution of the integral equation

The main result of our study is the following:
**Proposition 1.** The expression

$$q(\varepsilon) = -\frac{f^+(\varepsilon)f^-(\varepsilon)}{2R(\varepsilon)} - \frac{\gamma(2\gamma - \mathcal{G}\hat{\Delta}_x)f^-(\varepsilon)}{f^+(\varepsilon)R(\varepsilon)} - \frac{\lambda(2\lambda - \mathcal{G}\hat{\Delta}_y)f^+(\varepsilon)}{f^-(\varepsilon)R(\varepsilon)}$$
$$- \frac{\mathcal{G}}{2}\int_{\omega_0}^{\omega} du\, \rho(u) \frac{1}{\varepsilon - u} \frac{f^+(\varepsilon)f^-(\varepsilon)R(u) - f^+(u)f^-(u)R(\varepsilon)}{R(\varepsilon)} \tag{16}$$

is a solution to the integral equation (13) with the 'gap' equations (14) satisfied. Furthermore the continuum limit of (6),

$$E = L\int_{\omega_0}^{\omega} d\varepsilon\, f^+(\varepsilon)f^-(\varepsilon)q(\varepsilon),$$

corresponds to the ground-state energy of the model.
**Proof:** We set

$$B^-(\varepsilon) = \frac{f^-(\varepsilon)^2}{R(\varepsilon)}, \quad B^+(\varepsilon) = \frac{f^+(\varepsilon)^2}{R(\varepsilon)}, \quad C(\varepsilon) = \int_{\omega_0}^{\omega} du\, \rho(u) \frac{f^0(\varepsilon) - f^0(u)}{\varepsilon - u},$$

$$D(\varepsilon) = \int_{\omega_0}^{\omega} du\, \rho(u) \frac{R(\varepsilon) - R(u)}{\varepsilon - u}, \quad f^0(\varepsilon) = \sqrt{\varepsilon^2 - \beta^2}.$$

The main challenge is to calculate the following term in (13),

$$
\mathcal{G} \int_{\omega_0}^{\omega} du\, \rho(u) f^0(u) \frac{q(\varepsilon) - q(u)}{\varepsilon - u}
$$

$$
= \mathcal{G} \int_{\omega_0}^{\omega} du\, \rho(u) \frac{f^0(u) q(\varepsilon) - f^0(\varepsilon) q(\varepsilon) + f^0(\varepsilon) q(\varepsilon) - f^0(u) q(u)}{\varepsilon - u}
$$

$$
= -\mathcal{G} q(\varepsilon) C(\varepsilon) + \mathcal{G} \int_{\omega_0}^{\omega} du\, \rho(u) \frac{f^0(\varepsilon) q(\varepsilon) - f^0(u) q(u)}{\varepsilon - u}.
$$

Rewriting the integral equation (13) as

$$
-\left( q(\varepsilon) + \frac{\mathcal{G} C(\varepsilon)}{2} \right)^2 + \frac{\mathcal{G}^2 C(\varepsilon)^2}{4} + \frac{1}{4} + \frac{\gamma^2}{f^+(\varepsilon)^2} + \frac{\lambda^2}{f^-(\varepsilon)^2}
$$

$$
+ \mathcal{G} \int_{\omega_0}^{\omega} du\, \rho(u) \frac{f^0(\varepsilon) q(\varepsilon) - f^0(u) q(u)}{\varepsilon - u} = 0, \quad (17)
$$

our task is to show that (17) holds. We adopt the following change of variables:

$$
\chi_x = 2\gamma - \mathcal{G} \hat{\Delta}_x, \qquad \chi_y = 2\lambda - \mathcal{G} \hat{\Delta}_y.
$$

That is, parameters $\chi_x, \chi_y$ replace $\gamma, \lambda$ in the following calculations. We begin with the square term in (17)

$$
\left( q(\varepsilon) + \frac{\mathcal{G}}{2} C(\varepsilon) \right)^2 = \frac{R(\varepsilon)^2}{4 f^0(\varepsilon)^2} + \frac{\mathcal{G}^2 \hat{\Delta}_x^2 \chi_x^2 f^-(\varepsilon)^2}{4 f^+(\varepsilon)^2 R(\varepsilon)^2} + \frac{\mathcal{G}^2 \hat{\Delta}_y^2 \chi_y^2 f^+(\varepsilon)^2}{4 f^-(\varepsilon)^2 R(\varepsilon)^2} + \frac{\mathcal{G}^2 f^0(\varepsilon)^2}{4 R(\varepsilon)^2} D(\varepsilon)^2
$$

$$
+ \frac{\mathcal{G} \hat{\Delta}_x \chi_x}{2 f^+(\varepsilon)^2} + \frac{\mathcal{G} \hat{\Delta}_y \chi_y}{2 f^-(\varepsilon)^2} + \frac{\mathcal{G}^2 \hat{\Delta}_x \hat{\Delta}_y \chi_x \chi_y}{2 R(\varepsilon)^2}
$$

$$
- \frac{\mathcal{G}}{2} D(\varepsilon) - \frac{\mathcal{G}^2 \hat{\Delta}_x \chi_x f^-(\varepsilon)^2}{2 R(\varepsilon)^2} D(\varepsilon) - \frac{\mathcal{G}^2 \hat{\Delta}_y \chi_y f^+(\varepsilon)^2}{2 R(\varepsilon)^2} D(\varepsilon). \quad (18)
$$

We then calculate the following term in (17)

$$
\mathcal{G} \int_{\omega_0}^{\omega} du\, \rho(u) \frac{f^0(\varepsilon) q(\varepsilon) - f^0(u) q(u)}{\varepsilon - u}
$$

$$
= -\frac{\mathcal{G}}{2} D(\varepsilon) - \frac{\mathcal{G}^2 \hat{\Delta}_x \chi_x}{2} \int_{\omega_0}^{\omega} du\, \rho(u) \frac{B^-(\varepsilon) - B^-(u)}{\varepsilon - u} - \frac{\mathcal{G}^2 \hat{\Delta}_y \chi_y}{2} \int_{\omega_0}^{\omega} du\, \rho(u) \frac{B^+(\varepsilon) - B^+(u)}{\varepsilon - u}
$$

$$
+ \frac{\mathcal{G}^2}{2} \int_{\omega_0}^{\omega} du\, \rho(u) \frac{\frac{f^0(\varepsilon)^2}{R(\varepsilon)} D(\varepsilon) - \frac{f^0(u)^2}{R(u)} D(u)}{\varepsilon - u} - \frac{\mathcal{G}^2}{2} \int_{\omega_0}^{\omega} du\, \rho(u) \frac{f^0(\varepsilon) C(\varepsilon) - f^0(u) C(u)}{\varepsilon - u}.
$$

$$
(19)
$$

Combining the gap equations (14), we set the following shorthand

$$
r = \int_{\omega_0}^{\omega} d\varepsilon\, \rho(\varepsilon) \frac{1}{R(\varepsilon)} = \frac{1}{2\beta} \left( \frac{\hat{\Delta}_x}{\chi_x} - \frac{\hat{\Delta}_y}{\chi_y} \right), \quad (20)
$$

$$
s = \int_{\omega_0}^{\omega} d\varepsilon\, \rho(\varepsilon) \frac{\varepsilon}{R(\varepsilon)} = -\frac{1}{2} \left( \frac{\hat{\Delta}_x}{\chi_x} + \frac{\hat{\Delta}_y}{\chi_y} \right). \quad (21)
$$

First,

$$
\int_{\omega_0}^{\omega} \mathrm{d}u \, \rho(u) \frac{B^-(\varepsilon) - B^-(u)}{\varepsilon - u}
$$
$$
= \int_{\omega_0}^{\omega} \mathrm{d}u \, \rho(u) \left[ \frac{\varepsilon R(u) - u R(\varepsilon) + \varepsilon R(\varepsilon) - \varepsilon R(\varepsilon)}{(\varepsilon - u) R(\varepsilon) R(u)} + \beta \frac{R(\varepsilon) - R(u)}{(\varepsilon - u) R(\varepsilon) R(u)} \right]
$$
$$
= r - f^-(\varepsilon)^2 \int_{\omega_0}^{\omega} \mathrm{d}u \, \rho(u) \frac{R(\varepsilon) - R(u)}{(\varepsilon - u) R(\varepsilon) R(u)}. \tag{22}
$$

Likewise

$$
\int_{\omega_0}^{\omega} \mathrm{d}u \, \rho(u) \frac{B^+(\varepsilon) - B^+(u)}{\varepsilon - u} = r - f^+(\varepsilon)^2 \int_{\omega_0}^{\omega} \mathrm{d}u \, \rho(u) \frac{R(\varepsilon) - R(u)}{(\varepsilon - u) R(\varepsilon) R(u)}. \tag{23}
$$

Hence

$$
- \frac{\mathcal{G}^2 \hat{\Delta}_x \chi_x}{2} \int_{\omega_0}^{\omega} \mathrm{d}u \, \rho(u) \frac{B^-(\varepsilon) - B^-(u)}{\varepsilon - u} - \frac{\mathcal{G}^2 \hat{\Delta}_y \chi_y}{2} \int_{\omega_0}^{\omega} \mathrm{d}u \, \rho(u) \frac{B^+(\varepsilon) - B^+(u)}{\varepsilon - u}
$$
$$
= - \frac{r \mathcal{G}^2}{2} (\hat{\Delta}_x \chi_x + \hat{\Delta}_y \chi_y) + \frac{\mathcal{G}^2 \hat{\Delta}_x \chi_x f^-(\varepsilon)^2}{2 R(\varepsilon)} \int_{\omega_0}^{\omega} \mathrm{d}u \, \rho(u) \frac{R(\varepsilon) - R(u)}{(\varepsilon - u) R(u)}
$$
$$
+ \frac{\mathcal{G}^2 \hat{\Delta}_y \chi_y f^+(\varepsilon)^2}{2 R(\varepsilon)} \int_{\omega_0}^{\omega} \mathrm{d}u \, \rho(u) \frac{R(\varepsilon) - R(u)}{(\varepsilon - u) R(u)}.
$$

Next

$$
\int_{\omega_0}^{\omega} \mathrm{d}u \, \rho(u) \frac{\frac{f^0(\varepsilon)^2}{R(\varepsilon)} D(\varepsilon) - \frac{f^0(u)^2}{R(u)} D(u)}{\varepsilon - u}
$$
$$
= \int_{\omega_0}^{\omega} \mathrm{d}u \, \rho(u) \frac{D(u)}{R(u)} (\varepsilon + u) + f^0(\varepsilon)^2 \int_{\omega_0}^{\omega} \mathrm{d}u \, \rho(u) \frac{R(u) D(\varepsilon) - R(\varepsilon) D(u)}{(\varepsilon - u) R(\varepsilon) R(u)}. \tag{24}
$$

By exploiting symmetries, the first term in (24) can be reduced as

$$
\int_{\omega_0}^{\omega} \mathrm{d}u \, \rho(u) \frac{D(u)}{R(u)} (\varepsilon + u)
$$
$$
= \frac{\varepsilon}{2} \int_{\omega_0}^{\omega} \mathrm{d}u \, \rho(u) \int_{\omega_0}^{\omega} \mathrm{d}v \, \rho(v) \frac{R(u) - R(v)}{u - v} \left( \frac{1}{R(u)} + \frac{1}{R(v)} \right)
$$
$$
+ \frac{1}{2} \int_{\omega_0}^{\omega} \mathrm{d}u \, \rho(u) \int_{\omega_0}^{\omega} \mathrm{d}v \, \rho(v) \frac{R(u) - R(v)}{u - v} \left( \frac{u}{R(u)} + \frac{v}{R(v)} \right)
$$
$$
= \frac{\varepsilon}{2} \int_{\omega_0}^{\omega} \mathrm{d}u \, \rho(u) \int_{\omega_0}^{\omega} \mathrm{d}v \, \rho(v) \frac{u^2 - v^2 + (\chi_x^2 + \chi_y^2)(u - v)}{(u - v) R(u) R(v)}
$$
$$
+ \frac{1}{2} \int_{\omega_0}^{\omega} \mathrm{d}u \, \rho(u) \int_{\omega_0}^{\omega} \mathrm{d}v \, \rho(v) \frac{u R(u) R(v) + v R(u)^2 - u R(v)^2 - v R(u) R(v)}{(u - v) R(u) R(v)}
$$
$$
= r s \varepsilon + \frac{r^2 \varepsilon}{2} \left( \chi_x^2 + \chi_y^2 \right) + \frac{1}{2} \left[ 1 + s^2 + (\chi_x^2 - \chi_y^2) \beta r^2 + \beta^2 r^2 \right], \tag{25}
$$

where we used the gap equations (14) in the last step.

Using a similar approach, we compute the second term in (24),

$$
\begin{aligned}
f^0(\varepsilon)^2 & \int_{\omega_0}^{\omega} du\, \rho(u) \frac{R(u)D(\varepsilon) - R(\varepsilon)D(u)}{(\varepsilon - u)R(\varepsilon)R(u)} \\
={}& \frac{f^0(\varepsilon)^2}{2R(\varepsilon)^2} \left( \int_{\omega_0}^{\omega} du\, \rho(u) \frac{R(\varepsilon) - R(u)}{\varepsilon - u} \right)^2 \\
& + \frac{f^0(\varepsilon)^2}{2} \int \int \left[ \frac{-\varepsilon R(u)^2 + u R(u)^2 + \varepsilon R(v)^2 - v R(v)^2}{(u-v)(\varepsilon-u)(\varepsilon-v)R(u)R(v)} - \frac{R(u)R(v)}{(\varepsilon-u)(\varepsilon-v)R(\varepsilon)^2} \right] \\
={}& \frac{f^0(\varepsilon)^2}{2R(\varepsilon)^2} D(\varepsilon)^2 - \frac{r^2 f^0(\varepsilon)^2}{2R(\varepsilon)^2} \left( (\chi_x^2 + \chi_y^2)^2 + (\chi_x^2 - \chi_y^2)\beta + \beta^2 + (\chi_x^2 + \chi_y^2)\varepsilon \right) \\
& - \frac{rs f^0(\varepsilon)^2}{R(\varepsilon)^2} \left( \chi_x^2 + \chi_y^2 + \varepsilon \right) - \frac{s^2 f^0(\varepsilon)^2}{2R(\varepsilon)^2},
\end{aligned}
\tag{26}
$$

where the shorthand "$\int \int$" for "$\int_{\omega_0}^{\omega} du\, \rho(u) \int_{\omega_0}^{\omega} dv\, \rho(v)$" is adopted and the last step is assisted by symbolic calculation in Mathematica.

Similarly, it can be shown that

$$
\begin{aligned}
\frac{\mathcal{G}^2}{2} & \int_{\omega_0}^{\omega} du\, \rho(u) \frac{f^0(\varepsilon)C(\varepsilon) - f^0(u)C(u)}{\varepsilon - u} \\
={}& \frac{\mathcal{G}^2}{4} \int \int \left[ \frac{(f^0(\varepsilon) - f^0(u))(f^0(\varepsilon) - f^0(v))}{(\varepsilon - u)(\varepsilon - v)} + \frac{f^0(\varepsilon)^2 - f^0(u)f^0(v)}{(\varepsilon - u)(\varepsilon - v)} \right. \\
& \left. - \frac{\varepsilon(f^0(u)^2 - f^0(v)^2)}{(\varepsilon - u)(\varepsilon - v)(u - v)} + \frac{v f^0(u)^2 - u f^0(v)^2 + f^0(u)f^0(v)(u - v)}{(\varepsilon - u)(\varepsilon - v)(u - v)} \right] \\
={}& \frac{\mathcal{G}^2}{4} C(\varepsilon)^2 + \frac{\mathcal{G}^2}{4}.
\end{aligned}
\tag{27}
$$

Before proceeding, we set

$$
\begin{aligned}
\mathcal{C}_1 ={}& -\frac{R(\varepsilon)^2}{4f^0(\varepsilon)^2} - \frac{\mathcal{G}^2 \hat{\Delta}_x^2 \chi_x^2 f^-(\varepsilon)^2}{4f^+(\varepsilon)^2 R(\varepsilon)^2} - \frac{\mathcal{G}^2 \hat{\Delta}_y^2 \chi_y^2 f^+(\varepsilon)^2}{4f^-(\varepsilon)^2 R(\varepsilon)^2} - \frac{\mathcal{G}\hat{\Delta}_x \chi_x}{2f^+(\varepsilon)^2} - \frac{\mathcal{G}\hat{\Delta}_y \chi_y}{2f^-(\varepsilon)^2} \\
& - \frac{\mathcal{G}^2 \hat{\Delta}_x \hat{\Delta}_y \chi_x \chi_y}{2R(\varepsilon)^2} + \frac{(\mathcal{G}\hat{\Delta}_x + \chi_x)^2}{4f^+(\varepsilon)^2} + \frac{(\mathcal{G}\hat{\Delta}_y + \chi_y)^2}{4f^-(\varepsilon)^2} + \frac{1}{4}, \\
\mathcal{C}_2 ={}& \frac{\mathcal{G}^2 \hat{\Delta}_x \chi_x f^-(\varepsilon)^2}{2R(\varepsilon)} \frac{D(\varepsilon)}{R(\varepsilon)} - \frac{\mathcal{G}^2 \hat{\Delta}_x \chi_x}{2} \int_{\omega_0}^{\omega} du\, \rho(u) \frac{B^-(\varepsilon) - B^-(u)}{\varepsilon - u}, \\
\mathcal{C}_3 ={}& \frac{\mathcal{G}^2 \hat{\Delta}_y \chi_y f^+(\varepsilon)^2}{2R(\varepsilon)} \frac{D(\varepsilon)}{R(\varepsilon)} - \frac{\mathcal{G}^2 \hat{\Delta}_y \chi_y}{2} \int_{\omega_0}^{\omega} du\, \rho(u) \frac{B^+(\varepsilon) - B^+(u)}{\varepsilon - u} \\
\mathcal{C}_4 ={}& \frac{\mathcal{G}^2}{2} \int_{\omega_0}^{\omega} du\, \rho(u) \frac{\dfrac{f^0(\varepsilon)^2}{R(\varepsilon)} D(\varepsilon) - \dfrac{f^0(u)^2}{R(u)} D(u)}{\varepsilon - u} - \frac{\mathcal{G}^2 f^0(\varepsilon)^2}{4R(\varepsilon)^2} D(\varepsilon)^2 \\
& + \frac{\mathcal{G}^2 C(\varepsilon)^2}{4} - \frac{\mathcal{G}^2}{2} \int_{\omega_0}^{\omega} du\, \rho(u) \frac{f^0(\varepsilon)C(\varepsilon) - f^0(u)C(u)}{\varepsilon - u}.
\end{aligned}
$$

Substituting (18) and (19) into (17), it remains to verify that

$$
\mathcal{C}_1 + \mathcal{C}_2 + \mathcal{C}_3 + \mathcal{C}_4 = 0.
\tag{28}
$$

From (22), we simplify $\mathcal{C}_2$ to obtain

$$
\mathcal{C}_2 = \frac{\mathcal{G}^2 \hat{\Delta}_x \chi_x f^-(\varepsilon)^2}{2R(\varepsilon)} \left( \frac{D(\varepsilon)}{R(\varepsilon)} + \int_{\omega_0}^{\omega} du\, \rho(u) \frac{R(\varepsilon) - R(u)}{(\varepsilon - u)R(u)} \right) - \frac{r\mathcal{G}^2 \hat{\Delta}_x \chi_x}{2}
$$

$$
= \frac{\mathcal{G}^2 \hat{\Delta}_x \chi_x f^-(\varepsilon)^2}{2R(\varepsilon)} \left[ \frac{r\varepsilon}{R(\varepsilon)} + \frac{r(\chi_x^2 + \chi_y^2) + s}{R(\varepsilon)} \right] - \frac{r\mathcal{G}^2 \hat{\Delta}_x \chi_x}{2}.
$$

Likewise, from (23) expression $\mathcal{C}_3$ can be reduced to

$$
\mathcal{C}_3 = \frac{\mathcal{G}^2 \hat{\Delta}_y \chi_y f^+(\varepsilon)^2}{2R(\varepsilon)} \left[ \frac{r\varepsilon}{R(\varepsilon)} + \frac{r(\chi_x^2 + \chi_y^2) + s}{R(\varepsilon)} \right] - \frac{r\mathcal{G}^2 \hat{\Delta}_y \chi_y}{2}.
$$

Substituting (25) and (26) into (24), and also from (27) expression $\mathcal{C}_4$ can be simplified as

$$
\mathcal{C}_4 = \frac{\mathcal{G}^2}{2} \left\{ rs\varepsilon + \frac{r^2\varepsilon}{2} \left( \chi_x^2 + \chi_y^2 \right) + \frac{1}{2} \left[ 1 + s^2 + (\chi_x^2 - \chi_y^2)\beta r^2 + \beta^2 r^2 \right] \right.
$$

$$
- \frac{r^2 f^0(\varepsilon)^2}{2R(\varepsilon)^2} \left[ (\chi_x^2 + \chi_y^2)^2 + (\chi_x^2 - \chi_y^2)\beta + \beta^2 + (\chi_x^2 + \chi_y^2)\varepsilon \right]
$$

$$
\left. - \frac{rs f^0(\varepsilon)^2}{R(\varepsilon)^2} \left( \chi_x^2 + \chi_y^2 + \varepsilon \right) - \frac{s^2 f^0(\varepsilon)^2}{2R(\varepsilon)^2} \right\} - \frac{\mathcal{G}^2}{4}.
$$

Lastly, using (20) and (21), it can be shown that (28) holds, thus concluding our proof that $q(\varepsilon)$ given by (16) is a solution to the integral equation (13).

Next, using the gap equations (14), the energy expression is calculated from (6) in the following

$$
E = \int_{\omega_0}^{\omega} d\varepsilon\, \rho(\varepsilon) f^+(\varepsilon) f^-(\varepsilon) q(\varepsilon)
$$

$$
= \frac{\mathcal{G}(\hat{\Delta}_x)^2}{2} + \frac{\mathcal{G}(\hat{\Delta}_y)^2}{2} - \frac{1}{2} \int_{\omega_0}^{\omega} d\varepsilon\, \rho(\varepsilon) R(\varepsilon)
$$

$$
- \frac{\mathcal{G}}{2} \int_{\omega_0}^{\omega} \int_{\omega_0}^{\omega} d\varepsilon\, du\, \rho(u)\rho(\varepsilon) \frac{f^+(\varepsilon)f^-(\varepsilon)}{R(\varepsilon)} \frac{f^+(\varepsilon)f^-(\varepsilon)R(u) - f^+(u)f^-(u)R(\varepsilon)}{\varepsilon - u}.
$$

Since

$$
- \frac{\mathcal{G}}{2} \int_{\omega_0}^{\omega} \int_{\omega_0}^{\omega} d\varepsilon\, du\, \rho(u)\rho(\varepsilon) \frac{f^+(\varepsilon)f^-(\varepsilon)}{R(\varepsilon)} \frac{f^+(\varepsilon)f^-(\varepsilon)R(u) - f^+(u)f^-(u)R(\varepsilon)}{\varepsilon - u}
$$

$$
= - \frac{\mathcal{G}}{4} \int_{\omega_0}^{\omega} \int_{\omega_0}^{\omega} d\varepsilon\, du\, \rho(u)\rho(\varepsilon) \frac{f^+(\varepsilon)f^-(\varepsilon)R(u) + f^+(u)f^-(u)R(\varepsilon)}{R(\varepsilon)R(u)}.
$$

$$
\cdot \frac{f^+(\varepsilon)f^-(\varepsilon)R(u) - f^+(u)f^-(u)R(\varepsilon)}{\varepsilon - u}
$$

$$
= - \frac{\mathcal{G}}{4} \left( (\hat{\Delta}_x)^2 + (\hat{\Delta}_y)^2 \right),
$$

we obtain the energy expression as

$$
E = \int_{\omega_0}^{\omega} d\varepsilon\, \rho(\varepsilon) f^+(\varepsilon) f^-(\varepsilon) q(\varepsilon) = \frac{\mathcal{G}(\hat{\Delta}_x)^2}{4} + \frac{\mathcal{G}(\hat{\Delta}_y)^2}{4} - \frac{1}{2} \int_{\omega_0}^{\omega} d\varepsilon\, \rho(\varepsilon) R(\varepsilon).
$$

Hence the integral approximation of the conserved operators $Q_i$ leads, through use of (15), to the same ground-state energy as that from the mean-field result (9). Q.E.D.

This result is a generalized solution to a similar problem in the continuum limit for the open $p + ip$ model [28].

## 5   Conclusion

In this study, we derived the ground-state energy of the generalized BCS Richardson-Gaudin model in the continuum limit through the COE. This was achieved by finding a solution to the integral equation derived from the quadratic relations of the COE. The solution corresponds to the ground state as its energy expression is consistent with a mean-field analysis. This method was originally established for the closed and open $p + ip$ models [28] in order to circumvent the difficulties in finding a complex curve to approximate the distribution of the roots of the BAE. For future work, it would be natural to derive the BAE for the generalized model and establish the connection between the BAE and the quadratic relations of the COE. In order to find the Bethe Ansatz solution to the generalized model, the methods of [10, 19, 20] could be extended to derive the solution. Alternatively, an extension of the method using polynomial equations, adopted for the closed [24] and open [11] models, could be developed to derive the BAE from the quadratic equations of the COE.

Setting $\beta = 0$ in (5), the method in [11] will suffice to derive the BAE as the quadratic relations for the COE coincide with the open case. The challenge, in the case when $\beta \neq 0$, is to generalize the previous method using polynomials to accommodate the quadratic term in (5),

$$g^2 \beta^2 \sum_{j \neq i}^{L} \left( \frac{1}{f_i^+ f_j^- + f_i^- f_j^+} \right)^2,$$

and $f_i^+ f_i^-$ generalizing $z_i^2$. Once the BAE are derived, the Bethe roots can be studied numerically and analytically. Furthermore, insights into topological properties of this model can be derived from both the mean-field analysis following methods adopted in [13], and the Bethe Ansatz solution following [14, 15].

## Acknowledgements

The authors acknowledge the traditional owners of the land on which The University of Queensland is situated, the Turrbal and Jagera people.

**Funding information**   This study was supported by the Australian Research Council through Discovery Project DP150101294.

## A   Mean-field calculation

Consider the spin-1/2 representation of the $su(2)$ operators

$$S^+ = \begin{pmatrix} 0 & 1 \\ 0 & 0 \end{pmatrix}, \quad S^- = \begin{pmatrix} 0 & 0 \\ 1 & 0 \end{pmatrix}, \quad S^z = \begin{pmatrix} 1/2 & 0 \\ 0 & -1/2 \end{pmatrix},$$
$$S^x = \begin{pmatrix} 0 & 1/2 \\ 1/2 & 0 \end{pmatrix}, \quad S^y = \begin{pmatrix} 0 & -i/2 \\ i/2 & 0 \end{pmatrix}.$$

Hence we have

$$
\begin{aligned}
H &= \sum_{i=1}^{L} f_i^+ f_i^- S_i^z + 2\gamma \mathcal{S}^x + 2\lambda \mathcal{S}^y - G\left(\mathcal{S}^x\right)^2 - G\left(\mathcal{S}^y\right)^2 + \frac{G}{4}\sum_{i=1}^{L}\left((f_i^-)^2 + (f_i^+)^2\right) \\
&\approx \sum_{i=1}^{L} f_i^+ f_i^- S_i^z + 2\gamma \mathcal{S}^x + 2\lambda \mathcal{S}^y - G\Delta_x \mathcal{S}^x - G\Delta_y \mathcal{S}^y + C \\
&= \frac{1}{2}\sum_{i=1}^{L}\begin{pmatrix} f_i^+ f_i^- & (2\gamma - G\Delta_x)f_i^- - i(2\lambda - G\Delta_y)f_i^+ \\ (2\gamma - G\Delta_x)f_i^- + i(2\lambda - G\Delta_y)f_i^+ & -f_i^+ f_i^- \end{pmatrix} + C,
\end{aligned}
$$

where

$$
C = \frac{G(\Delta_x)^2}{4} + \frac{G(\Delta_y)^2}{4} + \sum_{i=1}^{L}\frac{Gz_i^2}{2}.
$$

Now we consider the following eigenvalue problem,

$$
\frac{1}{2}\begin{pmatrix} f_i^+ f_i^- & (2\gamma - G\Delta_x)f_i^- - i(2\lambda - G\Delta_y)f_i^+ \\ (2\gamma - G\Delta_x)f_i^- + i(2\lambda - G\Delta_y)f_i^+ & -f_i^+ f_i^- \end{pmatrix}\bar{v}_i = \sigma_i \bar{v}_i,
$$

where

$$
\bar{v}_i = \begin{pmatrix} v_i \\ u_i \end{pmatrix},
$$

leading to

$$
\sigma_i^2 = \frac{(f_i^+)^2(f_i^-)^2 + (2\gamma - G\Delta_x)^2(f_i^-)^2 + (2\lambda - G\Delta_y)^2(f_i^+)^2}{4}.
$$

Minimising the energy of the Hamiltonian, we choose

$$
\sigma_i = -\frac{1}{2}\sqrt{(f_i^+)^2(f_i^-)^2 + (2\gamma - G\Delta_x)^2(f_i^-)^2 + (2\lambda - G\Delta_y)^2(f_i^+)^2}.
$$

The ground-state energy is given as

$$
\begin{aligned}
E = &-\frac{1}{2}\sum_{i=1}^{L}\sqrt{(f_i^+)^2(f_i^-)^2 + (2\gamma - G\Delta_x)^2(f_i^-)^2 + (2\lambda - G\Delta_y)^2(f_i^+)^2} \\
&+ \frac{G(\Delta_x)^2}{4} + \frac{G(\Delta_y)^2}{4} + \sum_{i=1}^{L}\frac{Gz_i^2}{2},
\end{aligned}
$$

and the ground-state reads

$$
|\Psi\rangle = \prod_{i=1}^{L}(u_i I + v_i S_i^+)|0\rangle.
$$

Let

$$
R_i = \sqrt{(f_i^+)^2(f_i^-)^2 + (2\gamma - G\Delta_x)^2(f_i^-)^2 + (2\lambda - G\Delta_y)^2(f_i^+)^2},
$$

we have

$$
\langle S_i^x \rangle = \frac{1}{2}(v_i^* u_i + u_i^* v_i) = \frac{(G\Delta_x - 2\gamma)f_i^-}{2R_i},
$$

$$
\langle S_i^y \rangle = \frac{i}{2}(-v_i^* u_i + u_i^* v_i) = \frac{(G\Delta_y - 2\lambda)f_i^+}{2R_i},
$$

$$
\langle S_i^z \rangle = \frac{1}{2}(|v_i|^2 - |u_i|^2) = \frac{f_i^+ f_i^-}{-2R_i}.
$$

Now we apply the Hellmann–Feynman theorem,

$$\left\langle \frac{\partial H}{\partial G} \right\rangle = -\frac{1}{2}(\Delta_x)^2 - \frac{1}{2}(\Delta_y)^2 + \frac{\partial C}{\partial G},$$

$$\frac{\partial E}{\partial G} = -\frac{1}{2}\sum_{i=1}^{L} \frac{-(2\gamma - G\Delta_x)\Delta_x(f_i^-)^2 - (2\lambda - G\Delta_y)\Delta_y(f_i^+)^2}{\sqrt{(f_i^+)^2(f_i^-)^2 + (2\gamma - G\Delta_x)^2(f_i^-)^2 + (2\lambda - G\Delta_y)^2(f_i^+)^2}} + \frac{\partial C}{\partial G},$$

leading to the equation

$$(\Delta_x)^2 + (\Delta_y)^2 = -\sum_{i=1}^{L} \frac{(2\gamma - G\Delta_x)\Delta_x(f_i^-)^2 + (2\lambda - G\Delta_y)\Delta_y(f_i^+)^2}{\sqrt{(f_i^+)^2(f_i^-)^2 + (2\gamma - G\Delta_x)^2(f_i^-)^2 + (2\lambda - G\Delta_y)^2(f_i^+)^2}}. \tag{29}$$

Next,

$$\left\langle \frac{\partial H}{\partial \gamma} \right\rangle = \Delta_x + \frac{\partial C}{\partial \gamma},$$

$$\frac{\partial E}{\partial \gamma} = -\sum_{i=1}^{L} \frac{(2\gamma - G\Delta_x)(f_i^-)^2}{\sqrt{(f_i^+)^2(f_i^-)^2 + (2\gamma - G\Delta_x)^2(f_i^-)^2 + (2\lambda - G\Delta_y)^2(f_i^+)^2}} + \frac{\partial C}{\partial \gamma},$$

leading to (10). Equation (11) can be derived similarly. It is obvious to see that the expressions for $\Delta_x$ in (10) and $\Delta_y$ in (11) lead to (29).

The mean-field results give the expectation value of the conserved operator (4) as

$$\langle Q_i \rangle = \langle S_i^z \rangle + \frac{2\gamma}{f_i^+}\langle S_i^x \rangle + \frac{2\lambda}{f_i^-}\langle S_i^y \rangle - 2G\sum_{j\neq i}^{L} \frac{1}{z_i^2 - z_j^2}\left(f_i^+ f_j^- \langle S_i^x \rangle\langle S_j^x \rangle + f_i^- f_j^+ \langle S_i^y \rangle\langle S_j^y \rangle\right)$$

$$- 2G\sum_{j\neq i}^{L} \frac{f_j^+ f_j^-}{z_i^2 - z_j^2}\left(\langle S_i^z \rangle\langle S_j^z \rangle - \frac{1}{4}\right)$$

$$= -\frac{f_i^+ f_i^-}{2R_i} - \frac{\gamma(2\gamma - G\Delta_x)f_i^-}{f_i^+ R_i} - \frac{\lambda(2\lambda - G\Delta_y)f_i^+}{f_i^- R_i} - \frac{G}{2}\sum_{j\neq i}^{L} \frac{f_i^+ f_i^- R_j - f_j^+ f_j^- R_i}{(z_i^2 - z_j^2)R_i}.$$

This expression is consistent with that of equation (16) in Proposition 1.

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
