# Peer review of "Ground-state energy of a Richardson-Gaudin integrable BCS model"

_SciPost Physics, doi:SciPost Phys. Core 2, 001 (2020)_

## Round 1 · Referee Report · Anonymous (Referee 2) · 2019-12-17

Strengths

1- Shows that the ground state's energy of a generalisation of the p+ip BCS pairing model can, in the continuum limit, be found exactly without specifying the Bethe equations and/or the Bethe roots, by making use of the quadratic equations linking the eigenvalues.

2- It proves the validity of the mean-field result in the description of the model's GS energy.

3- The submission is remarkably clear.

Weaknesses

1 - The paper is very close in spirit to previous work by the authors and, as such, does not necessarily introduce new techniques but simply a new application of a previously established approach.

2- The physical relevance of this particular generalisation of the BCS model is not discussed which might limit the appeal of the result to the "integrable-models community".

Report

The work presented in this submission is mathematically correct and presents a new result concerning the exact solution of a generalised p+ip BCS model.

The authors find the explicit ground state solution, in the continuum limit, of the quadratic equations characteristic of spin-1/2 Richardson-Gaudin models for the "non-skew symmetric"/"XYZ with field" model.

This demonstration provides an interesting new result further extending the usability of the method developed in their previous work (doi:10.1016/j.nuclphysb.2018.08.015).

This further proof of usability of a promising method should, in my opinion, warrant publication. I feel that the Expectations of Scipost physics (groundbreaking, etc) are not met by the submission and therefore recommend publication in SciPost physics Core.

Requested changes

No necessary changes. Could be published as is in Scipost Physics Core.

---

## Round 1 · Referee Report · Anonymous (Referee 3) · 2020-1-10

Strengths

1. Exact result for ground-state energy of an integrable system.
2. Detailed proof of result contained.

Weaknesses

1. No discussion of the physical interpretation of the model.
2. Limited scope of result.
3. Few results of link to previously studied systems.
4. The authors show that the obtained result is identical to the ground-state result in the mean-field treatment, but they should add an argument why this also implies that they have obtained the true ground-state energy.

Report

The authors derive the ground-state energy of a generalised BCS model and show that it agrees with the finding of a mean-field analysis. Overall the results seem sound and suitable for publication in SciPost Core once the requested changes have been addressed.

Requested changes

1. The model (1) should be discussed in terms of its physical interpretation. Also it should be clarified under which conditions the model simplifies to the previously studied ones (eg, p+ip model).
2. After (2) it is required that \beta>0, but after (5) the authors set \beta=0.
3. It is unclear what the term "open" refers to since I cannot identify anything like a heat or particle bath in the Hamiltonian (1).
4. The authors should extend the discussion of how the x_i introduced in Sec. 4 enter, eg, in (7). I am confused since I cannot see any notion of a length in (1). so I do not understand how the density is introduced.
5. Similarly in spirit, the authors should extend the discussion of how the q_i of (7) become the functions Q(\epsilon) in (13).
6. Clarify the conclusion "this solution corresponds to the ground state of the model" in Proposition 1 and its relation to the mean-field analysis.

---

## Round 2 · Author Response

We thank the referees for their helpful comments, which have led to improvements in the presentation of the results.

---

## Round 2 · List of Changes

Response to Anonymous Report 2.
While the referee did not specifically request changes, it was remarked that our work
\textit{does not necessarily introduce new techniques but simply a new application of a previously established approach}.
This comment showed us that a subtle aspect of our work was not clearly communicated in our first submission. Namely, that there are two gap equations associated to this model, in contrast to $s$-wave and $p+ip$ models which have one gap equation. This aspect did complicate the analysis, and in fact new techniques were required. We have highlighted this feature by adding the following text to the end of Sect. 3
\textit{Next we will introduce an integral approximation to show that the general mean-field ground-state energy (9) is consistent with the exact solution. The techniques used require an extension of those in [28], to account for the two gap equations.}
Response to Anonymous Report 3.
The referee requested that the following changes be made:
\begin{itemize}
\item[1.] The model (1) should be discussed in terms of its physical interpretation. Also it should be clarified under which conditions the model simplifies to the previously studied ones (eg, p+ip model).
\item[2.] After (2) it is required that $\beta>0$, but after (5) the authors set $\beta=0$.
\item[3.] It is unclear what the term "open" refers to since I cannot identify anything like a heat or particle bath in the Hamiltonian (1).
\item[4.] The authors should extend the discussion of how the $x_i$ introduced in Sec. 4 enter, eg, in (7). I am confused since I cannot see any notion of a length in (1). so I do not understand how the density is introduced.
\item[5.] Similarly in spirit, the authors should extend the discussion of how the $q_i$ of (7) become the functions $Q(\epsilon)$ in (13).
\item[6.] Clarify the conclusion "this solution corresponds to the ground state of the model" in Proposition 1 and its relation to the mean-field analysis.
\end{itemize}
We have implemented the following changes:
\begin{itemize}
\item[1.]
Text has been added regarding the physical interpretation of the model, viz
\textit{The Hamiltonian has the form of a standard BCS model, (the terms in the first line of (1)), with additional terms. The standard terms are associated with the single-particle energy spectrum and the scattering of Cooper pairs. The additional terms are responsible for the breaking of the $u(1)$-symmetry associated to the total particle number, and are interpreted as interaction with the system's environment. }
This appears on pages 3 and 4 of the revised version. The condition under which this model simplifies to the $p+ip$ model was stated in the original manuscript This statement, shown below, has been promoted to appear earlier in the manuscript, on page 4.
\textit{By setting $\lambda = 0$ and
\begin{align*}
\beta_x & = \beta_y \iff \beta =0,
\end{align*}
we recover the conserved operators underlying the open $p+ip$ model. Furthermore, if we also set $\gamma = 0$, we recover the closed $p+ip$ model. }
\item[2.] The strict inequality $\beta_x > \beta_y$ has been relaxed to $\beta_x \geq \beta_y$, such that $\beta=0$ is now possible.
\item[3.] To clarify this point, the original block of text
\textit{This extended model, named the ‘open’ model, includes an interaction term corresponding to particle exchange with the system’s environment, and thus no longer has conservation of particle number}
from the Introduction has been modified to read
\textit{This extended model no longer conserves particle number, and thus cannot be considered as describing a closed system. Therefore we refer to it as an open model. The interaction term corresponding to particle exchange with the system’s environment leads to ....}
\item[4.] The variable $x$ is not associated with the Hamiltonian. It is introduced to illustrate how the integral approximation is implemented. To help clarify this point, the text at the beginning of Sect. 4 has been modified to read \\
\textit{Recall that for an arbitrary function $F(x)$, we may consider the following integral approximation (or continuum limit) of a summation }
Moreover, the following text has been included on page 7
\textit{The Hamiltonian is a function of many independent variables, including $\{z_i\}$. It is physically plausible to interpret these particular variables as the momentum spectrum in the non-interacting limit with $\beta=0$. Consider a density $\rho$ for the distribution of the variables $\{z_i\}$. This density is associated with the kinetic energies in the non-interacting limit with $\beta=0$.}
\item[5.] We believe that this point has been clarified by the text above. Further, we have changed the notation $Q(\varepsilon)$ to $q(\varepsilon)$ throughout.
\item[6.] To improve clarity, the comment
\textit{Furthermore the continuum limit of (6),
\begin{align*}
E = L\int_{\omega_0}^\omega {\rm d}\varepsilon\,f^+(\varepsilon) f^-(\varepsilon) q(\varepsilon),
\end{align*}
corresponds to the ground-state energy of the model.}
has been included in the Proposition.
\end{itemize}
While the referee did not specifically request changes, it was remarked that our work
\textit{does not necessarily introduce new techniques but simply a new application of a previously established approach}.
This comment showed us that a subtle aspect of our work was not clearly communicated in our first submission. Namely, that there are two gap equations associated to this model, in contrast to $s$-wave and $p+ip$ models which have one gap equation. This aspect did complicate the analysis, and in fact new techniques were required. We have highlighted this feature by adding the following text to the end of Sect. 3
\textit{Next we will introduce an integral approximation to show that the general mean-field ground-state energy (9) is consistent with the exact solution. The techniques used require an extension of those in [28], to account for the two gap equations.}
Response to Anonymous Report 3.
The referee requested that the following changes be made:
\begin{itemize}
\item[1.] The model (1) should be discussed in terms of its physical interpretation. Also it should be clarified under which conditions the model simplifies to the previously studied ones (eg, p+ip model).
\item[2.] After (2) it is required that $\beta>0$, but after (5) the authors set $\beta=0$.
\item[3.] It is unclear what the term "open" refers to since I cannot identify anything like a heat or particle bath in the Hamiltonian (1).
\item[4.] The authors should extend the discussion of how the $x_i$ introduced in Sec. 4 enter, eg, in (7). I am confused since I cannot see any notion of a length in (1). so I do not understand how the density is introduced.
\item[5.] Similarly in spirit, the authors should extend the discussion of how the $q_i$ of (7) become the functions $Q(\epsilon)$ in (13).
\item[6.] Clarify the conclusion "this solution corresponds to the ground state of the model" in Proposition 1 and its relation to the mean-field analysis.
\end{itemize}
We have implemented the following changes:
\begin{itemize}
\item[1.]
Text has been added regarding the physical interpretation of the model, viz
\textit{The Hamiltonian has the form of a standard BCS model, (the terms in the first line of (1)), with additional terms. The standard terms are associated with the single-particle energy spectrum and the scattering of Cooper pairs. The additional terms are responsible for the breaking of the $u(1)$-symmetry associated to the total particle number, and are interpreted as interaction with the system's environment. }
This appears on pages 3 and 4 of the revised version. The condition under which this model simplifies to the $p+ip$ model was stated in the original manuscript This statement, shown below, has been promoted to appear earlier in the manuscript, on page 4.
\textit{By setting $\lambda = 0$ and
\begin{align*}
\beta_x & = \beta_y \iff \beta =0,
\end{align*}
we recover the conserved operators underlying the open $p+ip$ model. Furthermore, if we also set $\gamma = 0$, we recover the closed $p+ip$ model. }
\item[2.] The strict inequality $\beta_x > \beta_y$ has been relaxed to $\beta_x \geq \beta_y$, such that $\beta=0$ is now possible.
\item[3.] To clarify this point, the original block of text
\textit{This extended model, named the ‘open’ model, includes an interaction term corresponding to particle exchange with the system’s environment, and thus no longer has conservation of particle number}
from the Introduction has been modified to read
\textit{This extended model no longer conserves particle number, and thus cannot be considered as describing a closed system. Therefore we refer to it as an open model. The interaction term corresponding to particle exchange with the system’s environment leads to ....}
\item[4.] The variable $x$ is not associated with the Hamiltonian. It is introduced to illustrate how the integral approximation is implemented. To help clarify this point, the text at the beginning of Sect. 4 has been modified to read \\
\textit{Recall that for an arbitrary function $F(x)$, we may consider the following integral approximation (or continuum limit) of a summation }
Moreover, the following text has been included on page 7
\textit{The Hamiltonian is a function of many independent variables, including $\{z_i\}$. It is physically plausible to interpret these particular variables as the momentum spectrum in the non-interacting limit with $\beta=0$. Consider a density $\rho$ for the distribution of the variables $\{z_i\}$. This density is associated with the kinetic energies in the non-interacting limit with $\beta=0$.}
\item[5.] We believe that this point has been clarified by the text above. Further, we have changed the notation $Q(\varepsilon)$ to $q(\varepsilon)$ throughout.
\item[6.] To improve clarity, the comment
\textit{Furthermore the continuum limit of (6),
\begin{align*}
E = L\int_{\omega_0}^\omega {\rm d}\varepsilon\,f^+(\varepsilon) f^-(\varepsilon) q(\varepsilon),
\end{align*}
corresponds to the ground-state energy of the model.}
has been included in the Proposition.
\end{itemize}

---

## Editorial Decision

published